

# Genetic architecture study of rheumatoid arthritis and juvenile idiopathic arthritis

Jun Jia[1,*], Junyi Li[2,*], Xueming Yao[2], YuHang Zhang[3], Xiaohao Yang[3], Ping Wang[2], Qianghua Xia[2], Hakon Hakonarson[4,5,6] and Jin Li[2]

[1] Department of Surgery of Foot and Ankle, Tianjin Hospital, Tianjin, China
[2] Department of Cell Biology, 2011 Collaborative Innovation Center of Tianjin for Medical Epigenetics, Tianjin Key Laboratory of Medical Epigenetics, Tianjin Medical University, Tianjin, China
[3] Tianjin University of Traditional Chinese Medicine, Tianjin, China
[4] Center for Applied Genomics, Children's Hospital of Philadelphia, Philadelphia, PA, United States of America
[5] Division of Human Genetics, Children's Hospital of Philadelphia, Philadelphia, PA, United States of America
[6] Department of Pediatrics, Perelman School of Medicine, University of Pennsylvania, Philadelphia, PA, United States of America
[*] These authors contributed equally to this work.

## ABSTRACT

**Background**. Rheumatoid arthritis and juvenile idiopathic arthritis are two types of autoimmune diseases with inflammation at the joints, occurring to adults and children respectively. There are phenotypic overlaps between these two types of diseases, despite the age difference in patient groups.

**Methods**. To systematically compare the genetic architecture of them, we conducted analyses at gene and pathway levels and constructed protein-protein-interaction network based on summary statistics of genome-wide association studies of these two diseases. We examined their difference and similarity at each level.

**Results**. We observed extensive overlap in significant SNPs and genes at the human leukocyte antigen region. In addition, several SNPs in other regions of the human genome were also significantly associated with both diseases. We found significantly associated genes enriched in 32 pathways shared by both diseases. Excluding genes in the human leukocyte antigen region, significant enrichment is present for pathways like interleukin-27 pathway and NO2-dependent interleukin-12 pathway in natural killer cells.

**Discussion**. The identification of commonly associated genes and pathways may help in finding population at risk for both diseases, as well as shed light on repositioning and designing drugs for both diseases.

Corresponding authors
Qianghua Xia, qhxia@tmu.edu.cn
Jin Li, jli01@tmu.edu.cn

## INTRODUCTION

Rheumatoid arthritis (RA) is a symmetric polyarticular arthritis that primarily affects the small diarthrodial joints of the hands and feet, while juvenile idiopathic arthritis (JIA) is caused by unknown etiology and persists at least 6 weeks in children under the age of

16, which does not contain other known conditions (*Firestein, 2003*; *Prakken, Albani & Martini, 2011*). The prevalence rate of RA varies from 0.41 to 0.54% from 2004 to 2014 based on US administrative health insurance claims databases (*Hunter et al., 2017*), which is observably greater than the prevalence rate of JIA ranging from 0.0038 to 0.40% according to a systematic literature review including 29 articles (*Thierry et al., 2014*).

Phenotypically, RA and JIA are similar in some aspects. They show some common symptoms and physical signs such as joint pain and swelling, limited joint mobility and deformity, morning stiffness, elevated rheumatoid factor, fever, etc. Some of the subtypes of JIA, such as polyarticular JIA, are particularly similar to RA. However, with distinct clinical and pathological features of these two diseases being noted, they have been defined as separate diseases by International League of Associations for Rheumatology (*Petty et al., 2004*). In particular, JIA is heterogeneous with variable clinical presentation and outcome. It has been classified into seven subtypes, including oligoarticular JIA (persistent/ extended), polyarticular JIA with negative rheumatoid factor (RF), polyarticular JIA with positive RF, psoriatic JIA, enthesitis related arthritis, systemic JIA and undifferentiated JIA (*Nigrovic, Raychaudhuri & Thompson, 2018*). RA is more homogeneous but with a poorer outcome.

It has long been recognized that both RA and JIA are related to autoimmune and inflammatory disorders (*Ravelli & Martini, 2007*; *Scott, Wolfe & Huizinga, 2010*). Population-based heritability estimates for RA and JIA are both 60% approximately (*Macgregor et al., 2000*; *Prahalad, 2006*). Single-nucleotide polymorphism (SNP)-based heritability for RA has been reported to be around 50% (*Speed & Balding, 2014*; *Speed et al., 2012*), slightly lower than that of JIA estimated to be 73% (*Li et al., 2015b*). Certain alleles in the HLA region are strong genetic predisposition factors for RA and JIA. It has been reported that for both RA and JIA, the odds ratio of HLA region is about 2.8, while that of most non-HLA loci is only 1.1 to 1.4. (*Nigrovic, Raychaudhuri & Thompson, 2018*; *Hersh & Prahalad, 2015*; *Viatte, Plant & Raychaudhuri, 2013*) The genetic predisposition of JIA is attributable to HLA class II molecules (HLA-DRB1, HLA-DPB1), HLA class I molecules and non-HLA genes. The clinical presentation of RF-positive JIA resembles that of RA, and they share the HLA-DRB1 epitope (*De Silvestri et al., 2017*; *Hinks et al., 2018*). The HLA-DRB1*04 confers a protective role in JIA before the age of 6, while it renders an increased risk of RA (*Nigrovic, Raychaudhuri & Thompson, 2018*). The immunopathogenesis of RA has become clear in recent years, but the pathogenesis of JIA remains unknown (*Firestein & McInnes, 2017*; *Mellins, Macaubas & Grom, 2011*).

With the rapid development of genomic technology, a large number of genetic variants associated with RA or JIA have been identified. To date, genome-wide association studies (GWASs) have identified a large number of variants associated with RA and JIA respectively. A total of 789 RA-associated variants from 52 studies and 129 JIA-associated variants from 11 studies have been reported in GWAS Catalog (association testing $P$-value $<1 \times 10^{-5}$) (*Buniello et al., 2019*), including 101 loci associated with RA and around 30 loci associated with JIA at genome-wide significant level. We aimed to compare the genetic architecture of RA and JIA at multiple levels systematically.

In this study, we conducted gene, pathway and network analyses of RA and JIA using robust and computational efficient methods based on their summary GWAS statistics. We

compared genetic difference and similarity between RA and JIA, identified their shared genetic signature. Significant overlap in genes and pathways were observed between these two diseases.

## MATERIALS AND METHODS

### Data collection

RA genetic loci information came from GWAS summary statistics of a trans-ethnic study (*Okada et al., 2014*) including a total of 29,880 RA cases and 73,758 controls of European and Asian ancestries. Summary statistics was downloaded from GWAS catalog (https://www.ebi.ac.uk/gwas/) (*Buniello et al., 2019*). All RA patients met the RA diagnostic criteria established by the American College of Rheumatology in 1987 (*Arnett et al., 1988*), or were confirmed by a professional rheumatologist (*Okada et al., 2014*).

JIA genetic loci information came from two resources. First, summary statistics of our previous GWAS on JIA (*Finkel et al., 2016*) was included in the current study. Our JIA GWAS is composed of discovery and replication cohorts with 1166 JIA cases and 9500 unrelated controls of European ancestry totally. Summary statistics of meta-analysis on the discovery and replication cohorts were used in our current study. Second, JIA variants revealed in published GWASs (*Behrens et al., 2008*; *Cobb et al., 2014*; *Finkel et al., 2016*; *Hinks et al., 2009*; *Hinks et al., 2013*; *Li et al., 2015a*; *Ombrello et al., 2017*; *Thompson et al., 2012*) were extracted from GWAS catalog (*Buniello et al., 2019*).

### Gene-based association analysis

A gene-based association analysis for RA and JIA was performed using *fastBAT* method (*Bakshi et al., 2016*) implemented in GCTA v1.91.7 (*Yang et al., 2011*) respectively, based on GWAS summary statistics of RA or JIA and linkage disequilibrium (LD) information from EUR population in the 1000 Genomes Project (*The Genomes Project Consortium et al., 2015*). Each gene region was defined as its transcript region and 50kb upstream/downstream, and the threshold for LD pruning was set to $r^2$-value >0.9, following the default setting of *fastBAT*. The gene list of human genome used by *fastBAT* method contains 24765 annotated genes (*Bakshi et al., 2016*), thus the genome-wide significant threshold for gene based tests was set at $0.05/24765 = 2 \times 10^{-6}$. JIA SNPs in GWAS catalog was also mapped to genes according to its report (*Buniello et al., 2019*).

### Protein-protein interaction network and pathway enrichment analyses

Competitive pathway enrichment analysis and protein-protein interaction (PPI) network visualization analysis were both performed using GWAS summary-level data by GSA-SNP2 (https://sites.google.com/view/gsasnp2) (*Yoon et al., 2018*). The LD information in the European population from the 1000 Genomes Project (*The Genomes Project Consortium et al., 2015*) was used to reduce false positives by combining highly correlated adjoining genes. Each gene region was defined as its transcript region and 20 kb upstream/downstream, as the default setting of GSA-SNP2. Gene-set database used for pathway construction were *C2(curated gene sets)CP(canonical pathways)v5.2*, which is a collection of online pathway

databases such as BioCarta (http://software.broadinstitute.org/gsea/msigdb/genesets.jsp?collection=CP:BIOCARTA) (*Nishimura, 2001*), KEGG (https://www.genome.jp/kegg/) (*Kanehisa et al., 2017*; *Kanehisa & Goto, 2000*; *Kanehisa et al., 2019*), Reactome ( https://reactome.org/) (*Fabregat et al., 2018*) and PID (*Schaefer et al., 2009*) by Molecular Signatures Database (MSigDB) (http://software.broadinstitute.org/gsea/msigdb) (*Liberzon et al., 2015*; *Liberzon et al., 2011*; *Subramanian et al., 2005*). The network data resource was the STRING database (https://string-db.org/) (*Szklarczyk et al., 2015*). Multiple-testing adjustment was performed and Q-value < 0.05 was set as the significance threshold. Global visual networks were constructed at a threshold of gene-score < 0.005 and Q-value < 0.05.

# RESULTS

## SNP-level comparison

A total of 26,285 SNPs (Table S1) in RA study and 105 SNPs (Tables S2, S3) in JIA study reached genome-wide significance threshold $P$-value $<5 \times 10^{-8}$, and these two diseases shared 47 significant SNPs. Among these SNPs, 37 were located in the human leukocyte antigen (HLA) region on chromosome 6. The rest 10 SNPs were located in or close to 9 genes (Table 1). Interestingly, 8 SNPs located in the HLA region showed opposite direction of effects, which meant risk allele of JIA could be protective allele for RA and vice versa.

## Gene-based comparison

To increase statistical power and to consider the combined effects of SNPs in genes, we conducted gene-disease association analyses, based on SNP-level summary statistics and taking into account of LD between SNPs. Several methods have been developed for computing gene-level associations based on SNP-level summary statistics, such as the commonly used PLINK (*Purcell et al., 2007*) set-baesd test and software VEGAS (Versatile Gene-based Association Study) (*Liu et al., 2010*), which are permutation and simulation-based approaches respectively. Both methods rely on resampling which is computationally intensive. Here, we adopted the *fastBAT* method which was a robust set-based association test computing the $P$-value of a gene with a number of SNPs from an approximated distribution (*Bakshi et al., 2016*). 431 genes located at 50 loci reached genome-wide significance in the RA dataset, including 17 known loci (*Acosta-Herrera et al., 2019*; *Buniello et al., 2019*; *Eyre et al., 2012*; *Plenge et al., 2005*; *Raychaudhuri et al., 2009*; *Zhu et al., 2016*) and 33 novel loci which should be examined in future replication studies (Table S4).

However only genes in the HLA region showed genome-wide significant association with JIA, which was likely due to the limited power of our previous GWAS (Table S5). A total of 75 significant genes or regions in the HLA were shared by JIA and RA (Table S6). Then we checked whether significant genes in RA contained additional genome-wide significant SNPs in JIA reported in GWAS catalog. Not surprisingly, one RA significant gene in the HLA region and 8 genes outside the HLA region containing genome-wide significant SNPs for JIA (Table 2) were observed. Because the *fastBAT* method conducted LD-pruning before combining SNP statistics, the top SNP showed in Table 2 may not be the one with the best $P$-value in original GWAS.

**Table 1  Genome-wide significant SNPs shared by RA and JIA (*P*-value < 5 × 10⁻⁸).** The raw data of genome-wide significant SNPs of RA are presented in Table S1; and the raw data of genome-wide significant SNPs of JIA are shown in Tables S2 and S3.

| SNP | Chr | Pos | RA | | | JIA | | | RefSeq gene |
|---|---|---|---|---|---|---|---|---|---|
| | | | Allele | OR | Pval | Allele | OR | Pval | |
| rs6679677 | 1 | 114303808 | A | 1.81 | 2.1E−149 | A | 1.59 | 3E−25 | 644bp 3′ of *RSBN1* |
| rs10174238 | 2 | 191973034 | G | 1.14 | 1.2E−13 | G | 1.29 | 1E−13 | *STAT4* |
| rs10213692 | 5 | 55442249 | T | 1.19 | 1.3E−17 | | 1.27 | 3E−11 | *ANKRD55* |
| rs7731626 | 5 | 55444683 | G | 1.20 | 7.3E−24 | A | | 1E−10 | *ANKRD55* |
| rs2517930 | 6 | 29745075 | T | 1.18 | 1.7E−31 | T | 1.47 | 8.95E−11 | 14kb 3′ of *HCG4* |
| rs2975033 | 6 | 29822261 | A | 1.18 | 1.6E−33 | A | 1.47 | 6.48E−10 | 23kb 3′ of *HLA-G* |
| rs12206499 | 6 | 29937127 | G | 1.16 | 6.4E−26 | G | 1.41 | 3.59E−08 | 5.8kb 5′ of *HCG9* |
| rs3823355 | 6 | 29942083 | T | 1.16 | 6.5E−26 | T | 1.43 | 1.10E−08 | 807bp 5′ of *HCG9* |
| rs6904029 | 6 | 29943067 | A | 1.16 | 6.8E−26 | A | 1.43 | 1.44E−08 | *HCG9* |
| rs3823375 | 6 | 29944158 | C | 1.16 | 1.7E−25 | C | 1.44 | 3.10E−09 | *HCG9* |
| rs9366752 | 6 | 30024677 | T | 1.09 | 1.6E−09 | T | 1.51 | 2.97E−10 | *ZNRD1-AS1* |
| rs1265048 | 6 | 31081409 | C | 1.12 | 5.3E−17 | C | 1.44 | 2.91E−09 | 1.1kb 5′ of *C6orf15* |
| rs13202464 | 6 | 31344583 | G | 1.19 | 1.5E−15 | G | 2.00 | 2.09E−11 | 20kb 5′ of *HLA-B* |
| rs9266689 | 6 | 31348580 | G | 1.14 | 3.3E−19 | G | 1.54 | 6.16E−11 | 19kb 5′ of *MICA* |
| rs2844533 | 6 | 31350802 | A | 1.30 | 6.6E−55 | A | 1.61 | 2.90E−08 | 17kb 5′ of *MICA* |
| rs2261033 | 6 | 31603591 | G | 1.56 | 4.2E−183 | G | 1.48 | 5.09E−09 | *PRRC2A* |
| rs6941112 | 6 | 31946614 | A | 1.31 | 6.1E−83 | A | 1.42 | 3.20E−09 | *STK19* |
| rs8111 | 6 | 32083175 | T | 1.33 | 7.2E−86 | T | 1.49 | 5.60E−11 | *ATF6B* |
| rs204999 | 6 | 32109979 | A | 1.55 | 5.5E−134 | A | 1.53 | 5.88E−09 | 6.2kb 3′ of *PRRT1* |
| rs17576984 | 6 | 32212985 | C | 1.54 | 3.0E−72 | T | 1.86 | 1.66E−12 | 21kb 5′ of *NOTCH4* |
| rs570963 | 6 | 32289594 | A | 1.18 | 2.9E−18 | G | 1.70 | 8.91E−11 | *C6orf10* |
| rs910049 | 6 | 32315727 | C | 1.19 | 5.2E−24 | C | 1.65 | 5.48E−10 | *C6orf10* |
| rs2395148 | 6 | 32321554 | G | 1.41 | 1.0E−20 | T | 3.62 | 1.08E−25 | *C6orf10* |
| rs6907322 | 6 | 32324945 | G | 1.14 | 1.7E−15 | A | 1.69 | 9.99E−15 | *C6orf10* |
| rs9268365 | 6 | 32333439 | G | 1.16 | 1.3E−20 | T | 1.66 | 4.98E−14 | *C6orf10* |
| rs3129941 | 6 | 32337686 | G | 1.64 | 1.4E−133 | G | 1.60 | 1.48E−09 | *C6orf10* |
| rs41291794 | 6 | 32425762 | A | 1.65 | 1.1E−63 | | 2.10 | 4E−15 | 13kb 3′ of *HLA-DRA* |
| rs2395185 | 6 | 32433167 | T | 2.01 | 1.0E−250 | G | 1.81 | 1.19E−16 | 20kb 3′ of *HLA-DRA* |
| rs477515 | 6 | 32569691 | A | 1.99 | 1.0E−250 | G | 1.89 | 3.19E−18 | 12kb 5′ of *HLA-DRB1* |
| rs2516049 | 6 | 32570400 | C | 2.00 | 1.0E−250 | T | 1.89 | 2.62E−18 | 13kb 5′ of *HLA-DRB1* |
| rs2858870 | 6 | 32572251 | T | 1.86 | 1.1E−77 | T | 2.19 | 8.41E−12 | 15kb 5′ of *HLA-DRB1* |
| rs7775055 | 6 | 32657916 | C | 1.56 | 1.4E−60 | C | 6.01 | 3E−174 | 23kb 5′ of *HLA-DQB1* |
| rs9275224 | 6 | 32659878 | G | 2.13 | 1.0E−250 | G | 1.41 | 1.06E−08 | 25kb 5′ of *HLA-DQB1* |
| rs6457617 | 6 | 32663851 | T | 2.14 | 1.0E−250 | T | 1.40 | 1.10E−08 | 29kb 5′ of *HLA-DQB1* |
| rs2858308 | 6 | 32670000 | G | 1.61 | 7.8E−92 | G | 1.98 | 1.94E−08 | 36kb 5′ of *HLA-DQB1* |
| rs2856705 | 6 | 32670956 | C | 1.61 | 1.0E−91 | C | 1.99 | 1.64E−08 | 36kb 5′ of *HLA-DQB1* |
| rs13192471 | 6 | 32671103 | C | 1.49 | 4.8E−123 | C | 1.93 | 1.93E−19 | 37kb 5′ of *HLA-DQB1* |

**Table 1** (*continued*)

| SNP | Chr | Pos | RA | | | JIA | | | RefSeq gene |
|---|---|---|---|---|---|---|---|---|---|
| | | | Allele | OR | Pval | Allele | OR | Pval | |
| rs1794275 | 6 | 32671248 | A | 1.33 | 3.7E−69 | A | 1.82 | 3.47E−13 | 37kb 5′ of *HLA-DQB1* |
| rs7765379 | 6 | 32680928 | G | 1.89 | 1.0E−250 | G | 1.68 | 3.11E−10 | 28kb 5′ of *HLA-DQA2* |
| rs4713610 | 6 | 33107955 | G | 1.27 | 5.7E−49 | G | 1.54 | 7.54E−09 | 11kb 3′ of *HLA-DPB2* |
| rs9277912 | 6 | 33124658 | T | 1.26 | 1.1E−48 | T | 1.51 | 2.61E−08 | 5.8kb 3′ of *COL11A2* |
| rs706778 | 10 | 6098949 | T | 1.09 | 1.5E−10 | T | | 6E−09 | *IL2RA* |
| rs9532434 | 13 | 40355913 | C | 1.10 | 1.0E−11 | | 1.19 | 5E−08 | *COG6* |
| rs3825568 | 14 | 69260588 | T | 1.08 | 2.7E−08 | | 1.30 | 1E−08 | 802bp 5′ of *ZFP36L1* |
| rs2847293 | 18 | 12782448 | A | 1.12 | 1.2E−10 | A | 1.31 | 1E−12 | 3kb 3′ of *PTPN2* |
| rs34536443 | 19 | 10463118 | G | 1.46 | 4.4E−16 | | 1.79 | 1E−10 | *TYK2* |
| rs8129030 | 21 | 36712588 | A | 1.09 | 2.5E−09 | | 1.28 | 5E−09 | 291kb 5′ of *RUNX1* |

**Notes.**

SNP, single nucleotide polymorphism; Chr, chromosome; Pos, position on human genome build hg19 (NCBI GRCh37); RA, rheumatoid arthritis; JIA, juvenile idiopathic arthritis; Allele, risk allele; OR, odds ratio of risk allele; Pval, disease association P-value of risk SNP; RefSeq gene, the closest gene to each SNP and their relative positions based on Reference sequence (RefSeq) database (*O'Leary et al., 2016*).

**Table 2** **Genome-wide significant genes outside the HLA region shared by RA and JIA (gene-based *P*-value < 2 × 10⁻⁶).** The raw data of genome-wide significant genes of RA are shown in Table S4 and those of JIA are shown in Tables S3 and S5.

| Gene | Chr | Start-End | RA | | | JIA | |
|---|---|---|---|---|---|---|---|
| | | | Pval | TopSNP_Pval | TopSNP | TopSNP_Pval | TopSNP |
| *PHTF1* | 1 | 114239823-114301777 | 7.41E−43 | 1.7E−38 | rs1217416 | 3E−25 | rs6679677 |
| *RSBN1* | 1 | 114304453-114355070 | 2.08E−19 | 2.8E−35 | rs3811019 | 3E−25 | rs6679677 |
| *ANKRD55* | 5 | 55395506-55529186 | 2.42E−09 | 7.3E−24 | rs7731626 | 3E−11 | rs10213692 |
| *IL2RA* | 10 | 6052656-6104333 | 4.58E−07 | 1.5E−10 | rs706778 | 8E−10 | rs7909519 |
| *SUOX* | 12 | 56391042-56399309 | 6.95E−07 | 3.7E−07 | rs701006 | 4E−09 | rs1689510 |
| *LOC100996324* | 18 | 12739484-12749421 | 5.76E−11 | 3.4E−15 | rs2847297 | 1E−12 | rs2847293 |
| *PTPN2* | 18 | 12785476-12884334 | 9.99E−14 | 1.1E−15 | rs7241016 | 1E−12 | rs2847293 |
| *TYK2* | 19 | 10461203-10491248 | 4.02E−07 | 2.7E−06 | rs12459219 | 1E−10 | rs34536443 |

**Notes.**

Chr, chromosome; Start-End, start and end boundaries of the gene region on human genome build UCSC hg19 (NCBI GRCh37); RA, rheumatoid arthritis; JIA, juvenile idiopathic arthritis; Pval, gene-level P-value based on fastBAT method; TopSNP, the top associated GWAS SNP; TopSNP_Pval, smallest single-SNP GWAS P-value in the gene region.

## Pathway-level comparison

GWAS pathway analysis consider either competitive null hypothesis or self-contained null hypothesis. Many methods for GWAS pathway analysis have been developed, but they are still subjected to the issues of low power and being influenced by some free parameters. The recently developed GSA-SNP2 package (*Yoon et al., 2018*) uses the random set model to compute pathway enrichment with decent type I error control by integrating the gene scores adjusted by the number of SNPs mapped to each gene and removing high inter-gene correlated adjacent genes in each pathway. It does not require any key free parameters concurrently. We applied this method to our analyses. RA or JIA associated genes were enriched in numerous canonical pathways at a threshold of Q-value <0.05. A total of 32 enriched pathways were shared by RA and JIA, which mostly were immune-related pathways, such as allograft rejection, type 1 diabetes mellitus, graft versus host disease,

**Table 3   Enriched pathways shared by RA and JIA after loci in the HLA region being removed (Q- value < 0.05).**

| Pathway | Database | Size | RA | | | JIA | | |
|---|---|---|---|---|---|---|---|---|
| | | | Count | Pval | Qval | Count | Pval | Qval |
| TYPE I DIABETES MELLITUS | KEGG | 44 | 23 | 1.66E−07 | 1.58E−05 | 41 | 2.12E−05 | 0.001785 |
| IL27 PATHWAY | PID | 26 | 26 | 0.001158 | 0.027447 | 25 | 4.79E−08 | 8.63E−06 |
| NO2IL12 PATHWAY | BIOCARTA | 17 | 15 | 0.002316 | 0.047593 | 16 | 1.87E−05 | 0.001686 |

Notes.
Pathway, abbreviation for each enriched pathway; Database, database from which the pathways were extracted; Size, total number of genes in each pathway; RA, rheumatoid arthritis; JIA, juvenile idiopathic arthritis; Count, the number of RA/JIA- significant genes falling into each pathway; Pval, P-value of each pathway; Qval, Q-value of each pathway based on the trend curve adjusted gene scores.

antigen processing and presentation, autoimmune thyroid disease, asthma, etc. (Table S7). Most of these significant pathways were driven by genes in the HLA region. In order to explore the role of loci outside the HLA region for these two diseases, we performed pathway enrichment analysis again after removing loci in the HLA region based on their genomic coordinates. The HLA region was defined as chr6:28,477,797-33,448,354 (GRCh37/hg19). Pathways such as interleukin(IL)-27 pathway and NO2-dependent IL-12 pathway in natural killer (NK) cells were significantly enriched even after the HLA region loci were removed (Table 3). Global networks were visualized at a threshold of gene-score <0.005 (Figs. 1&2). We observed the common hub role of several genes such as *TYK2*. The networks before removing the HLA region were shown in Figs. S1 and S2.

## DISCUSSION

Despite the phenotypic similarity between JIA and RA, systematic comparison of genetic similarity and distinction between these two types of diseases are lacking. Large scale GWASs of RA and JIA respectively render us ability to conduct such comparison and to identify potential common mechanism in disease pathogenesis, which may help repositioning and designing treatment strategies.

To systematically compare the genetic architecture of the two diseases, we performed gene-level, pathway-level analyses and conducted comparison at each level. Not only did we observe a large amount of overlaps in the HLA region as expected, but we also observed several SNPs and genes which significantly associated with both diseases outside the HLA region. Among them, the risk alleles of several SNPs were different between the two diseases, which meant that a certain allele may play a risk role in one disease but a protective role in the other. These SNPs might be related to the differences in pathogenesis and phenotype between JIA and RA. As we did not perform genome-wide imputation analysis due to unavailability of individual-level data, the number of genome-wide significant SNPs shared by these two diseases was actually underestimated.

Due to the limited sample size of our JIA data, we could not perform analysis for each subtype of JIA with enough statistical power. However, the heterogeneity of JIA and the genetic basis of its subtypes are worth noting. Some HLA alleles show different directions of effects on different subtypes of JIA and RA. For instance, HLA-DRB1*8, HLA-DRB1*11 and HLA-DRB1*13 are risk alleles of seronegative JIA, but do not exhibit

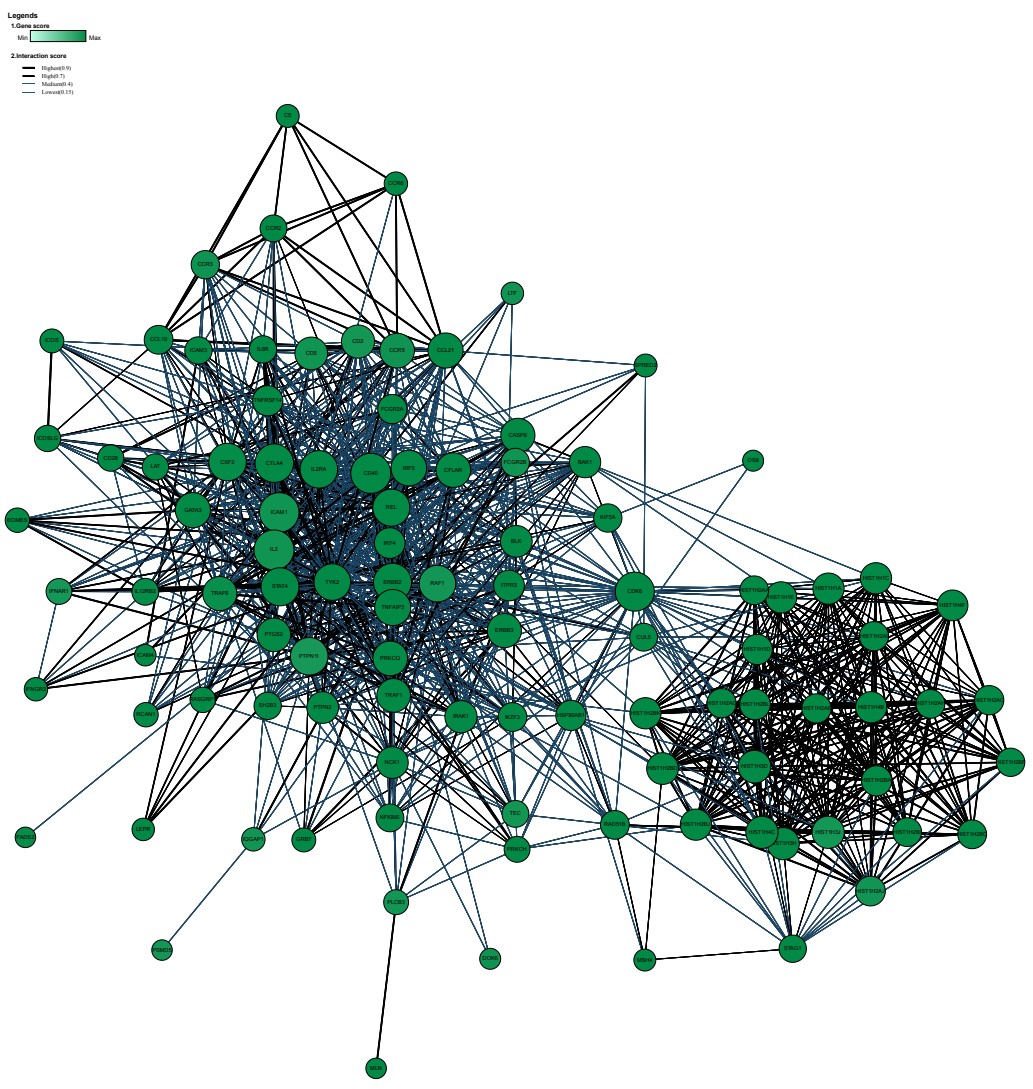

**Figure 1** **The global network of RA after the HLA region being removed (Q-value < 0.05, gene-score < 0.005).** The PPI network was constructed among proteins encoded by the significant RA-associated genes excluding those in the HLA region. The nodes in the figure represent the proteins and the connections between nodes indicate protein-protein interactions. The size of each node suggests the degrees of the connection between the node and the others.

association with seropositive polyarticular JIA and seronegative RA, and these HLA alleles render protective effect for seropositive RA. In particular, DRB1*11 is also a risk allele of systemic JIA, while the other two alleles are not associated with this JIA subtype (*Nigrovic, Raychaudhuri & Thompson, 2018*). As for alleles outside the HLA region, certain SNPs in genes *PTPN22* and *STAT1/STAT4* do not show association with systemic JIA, but confer risk for most other subtypes of JIA and RA (*Nigrovic, Raychaudhuri & Thompson, 2018*). In a recent study, Hinks et al. demonstrated that RF-positive polyarticular JIA is more similar to adult RA compared to other JIA subtypes in terms of genetic profile examined

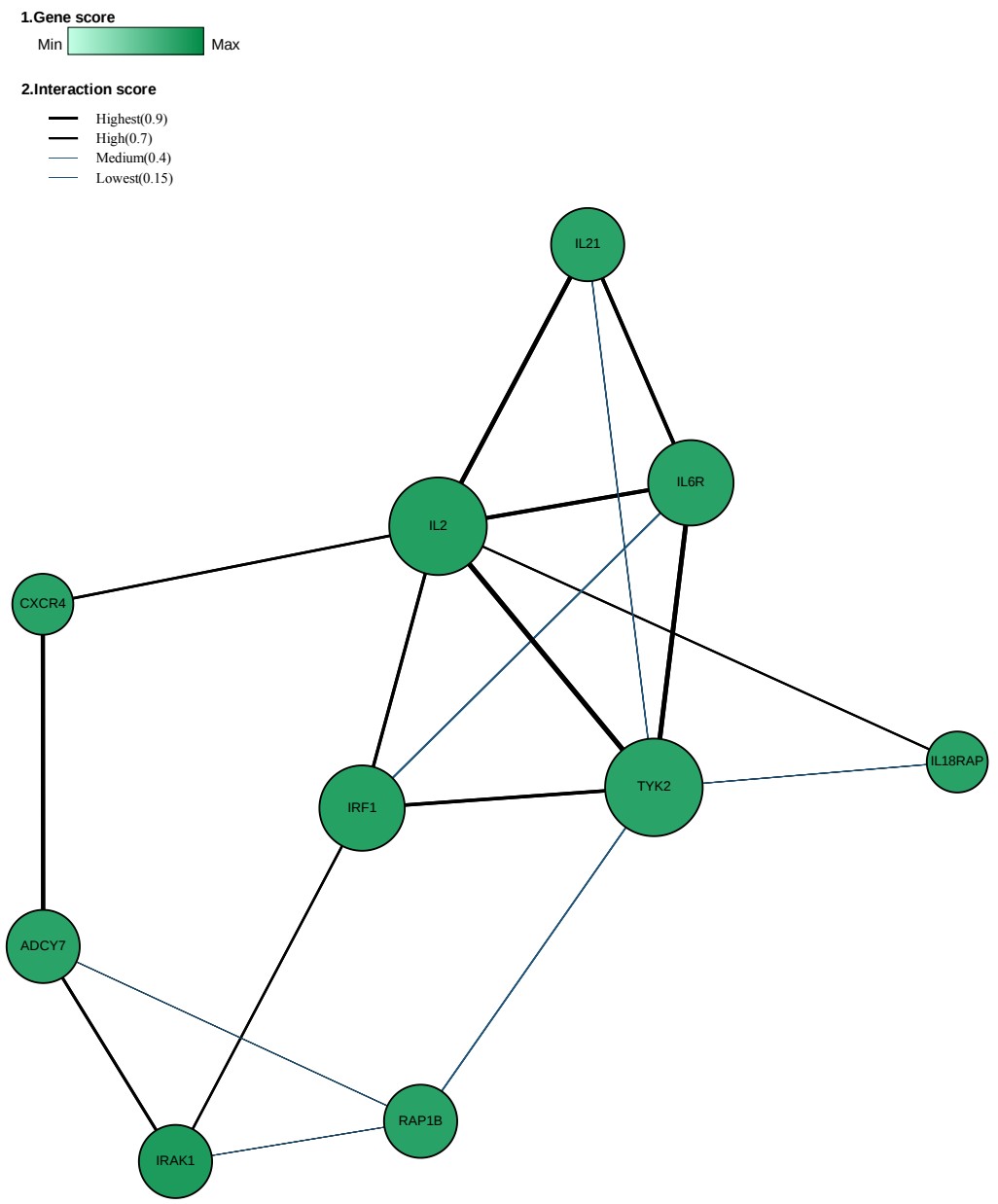

**Figure 2** **The global network of JIA after the HLA region being removed (Q-value < 0.05, gene-score < 0.005).** The PPI network was constructed among proteins encoded by the significant JIA-associated genes excluding those in the HLA region. The nodes in the figure represent the proteins and the connections between nodes indicate protein-protein interactions. The size of each node suggests the degrees of the connection between the node and the others.

on Immunochip (*Hinks et al., 2018*; *Onuora, 2018*). Further analysis of the genetic nature of different subtypes of JIA and RA would be helpful to optimize the classification of the two diseases, and may lead to more effective treatment and better prognosis.

We observed significant enrichment of *NO2-dependent IL12 pathway* and *IL27 pathway* for both RA and JIA. Macrophages release IL-12 which plays an important role in activation of NK cells and induces cytotoxicity with nitric oxide (*Liu et al., 2005*). NK cells are regarded as a bridge between innate and adaptive immunity, serving as a key regulator in the pathogenesis and development of autoimmune diseases (*Gianchecchi, Delfino & Fierabracci, 2018*). It has been reported that high percentages of NK cells and their activity were found in synovial fluid of active RA patients at advanced stage (*Yamin et al., 2019*), and dysfunction of NK cells was also observed in patients with systemic-onset JIA and its complication (*Grom et al., 2003*). NO2-dependent IL12 pathway plays a unique role in the activation of NK cells by macrophage. The enrichment of this pathway in our analyses implies the potential role of abnormal IL-12-mediated activation of NK cell in the pathogenesis of RA and JIA. IL-12 has long been considered as a therapeutic target of arthritis and other autoimmune and inflammatory disorders (*Hasko & Szabo, 1999*; *Siebert et al., 2015*). As a member of the IL-12 family, IL-27 induces T cell differentiation and causes immunosuppressive effects by inhibiting the development of Th17 cells (*Yoshida & Miyazaki, 2008*). Previous studies have suggested that IL-27 is another key modulator of autoimmunity and elevation of IL-27 signaling may be inhibitory to some autoimmune diseases, such as multiple sclerosis or uveitis (*Amadi-Obi et al., 2007*). Our results suggest that such therapeutic approach may be also applied to the management of RA and JIA.

## CONCLUSION

Our study identified genetic similarities and differences between RA and JIA at multiple levels. We observed a number of genes being associated with both diseases especially in the HLA region, and distinct genetic loci were found as well. Such systematic comparison and further functional characterization of these genetic loci and signaling pathways may lead to the identification of common drug targets for both diseases or drug repositioning, and may also contribute to the precision treatment of each disease.

### Funding

This study was supported by National Natural Science Foundation of China (No.81771769), Tianjin Natural Science Foundation (No.18JCYBJC42700), Startup Funding from Tianjin Medical University. The funders had no role in study design, data collection and analysis, decision to publish, or preparation of the manuscript.

### Grant Disclosures

The following grant information was disclosed by the authors:
National Natural Science Foundation of China: No.81771769.
Tianjin Natural Science Foundation: No.18JCYBJC42700.
Tianjin Medical University.

## Competing Interests

The authors declare there are no competing interests.

## Author Contributions

- Jun Jia and Junyi Li analyzed the data, performed the experiments, authored or reviewed drafts of the paper, and approved the final draft.
- Xueming Yao, YuHang Zhang, Xiaohao Yang and Ping Wang analyzed the data, prepared figures and/or tables, and approved the final draft.
- Qianghua Xia, Hakon Hakonarson and Jin Li conceived and designed the experiments, authored or reviewed drafts of the paper, and approved the final draft.

## Data Availability

All data generated in this study are available in the Supplemental Files.

## Supplemental Information

Supplemental information for this article can be found online at http://dx.doi.org/10.7717/peerj.8234#supplemental-information.

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
