# Peer review of "Genetic architecture study of rheumatoid arthritis and juvenile idiopathic arthritis"

_PeerJ, doi:10.7717/peerj.8234_

## Round 0.1 · original submission · Major Revisions

Please address the reviewer's comments.

Reviewer 1 ·

Basic reporting

To date, the largest study in the genetics of RA (Okada et al., Nature, 2014) has identified 101 loci associated to the disease. However, the authors report 789 loci. In the case of JIA, around 30 loci have been associated to the disease at the genome-wide significance level, but the authors report 129 loci. What is the origin of these discordant observations? This is an important point to clarify.

Experimental design

no comment

Validity of the findings

No comment

Additional comments

Jia et al. have performed a comparative analysis at gene- and pathway-levels between two related autoimmune conditions, rheumatoid arthritis (RA) and juvenile idiopathic arthritis (JIA). They have used recently developed methods for the study of complex traits. While applying the new methodologies is interesting, the work has some concerns that have to be addressed:

1. The authors performed a gene-based association analysis using fastBAT method and identified a huge numbers of genes -431- associated in the RA dataset. However, it seems that the authors do not pay special attention to these results: are the 431 genes identified already known genes for RA? Are there new genes for RA between them?

2. Some of the descriptions are presented ambiguously: in the abstract, the authors stated that ´we conducted analyses at the SNP, gene and pathway levels´; however, they actually did not perform the SNP-level analysis. This is an important issue that should be avoided through all the manuscript.

3. The last sentence of Introduction section says that the ‘results suggest that some genetic variants can significantly increase the susceptibility of both diseases’; however, the present work does not formally assess (through a statistical test) this question. The work is limited to a comparison between SNPs and genes for RA and JIA. Therefore, the author should be more rigorous when reporting the conclusions from their actual approach and results.

Minor point: there are some typos on lines 164-165, 192-193.

Reviewer 2 ·

Basic reporting

It is an interesting article that needs some more clinical and biological concepts, in order to make it more practical.

Experimental design

The experimental design is clear, but I suggested the authors to retrieve some more data if possible.

Validity of the findings

no comment

Additional comments

The authors addressed several issues I had previously raised, but the manuscript still needs significant changes, although it has been improved by the authors. Moreover, the inclusion of two additional cases increased the scientific value of this paper.

INTRODUCTION
- in the first two paragraphs, although the JIA definition is appropriate, it’s important to provide the reader (who may not be an experienced pediatric rheumatologist) with some additional information about JIA, especially as regards the classification in 7 forms according to EULAR classification, which is quite different from AR or, more in general, chronic arthritis in adults.

-some concepts about genetic predisposition to JIA and AR are completely lacking in the introduction, especially as regards HLA association. I recommend the authors to fill this gap by taking advantage of some specific references that consider and discuss this aspect under the double point of view of JIA and AR (HLA-DRB1 alleles and juvenile idiopathic arthritis: Diagnostic clues emerging from a meta-analysis. Autoimmun Rev. 2017 Dec;16(12):1230-1236. doi: 10.1016/j.autrev.2017.10.007; & Genetics and the Classification of Arthritis in Adults and Children. Arthritis Rheumatol. 2018 Jan;70(1):7-17. doi: 10.1002/art.40350)

MATERIALS and METHODS/RESULTS

-did the authors perform their research taking in account also the different subtypes of chronic arthritis, especially as regards JIA (with its different subtypes)? Indeed, systemic forms are increasingly considered autoinflammatory syndromes (as well as AOSD for AR) and some specific forms (e.g. psoriatic) are considered separately from AR in adults. If they have not done that, is it possible to retrieve this data as well? If so, how are the results? Anyway, this is a point that deserve some discussion whatever is the answer or results, as emphasized also by the references recommended above.

DISCUSSION
-in addition to address the observation right above, I would suggest the authors to provide some immunological concepts about their main observation (“We observed significant enrichment of NO2-dependent IL12 pathway and IL27 pathway for both RA and JIA”). It would be important to emphasize the potential biological (and practical) significance of their research.
-please, separate discussion and conclusion.

---

## Round 0.2 · Minor Revisions

Please address the minor comments mentioned by the reviewers.

Reviewer 1 ·

Basic reporting

no comment

Experimental design

no comment

Validity of the findings

In response to:

'Response: We agree with the reviewer and have revised our manuscript accordingly.
The revised introduction section: “Our results suggest that some genetic variants are associated with both diseases and their unique genetic risk loci may be related to their distinct clinical features .” ' -->

Again, they are not doing any SNP association test. They did not discover any new associated variant to both diseases. Please limit the conclusions to the actual results of the manuscript.

Reviewer 2 ·

Basic reporting

The article is well-written: there are a few minor grammar inconsistencies that the authors can easily fix by proof correction. Tables, figures and references are appropriate.

Experimental design

Experimental design and aims are clearly explained.

Validity of the findings

Both the discussion and conclusion improved with authors' revision. I have no additional major concerns.

Additional comments

The authors addressed appropriately my main observations.

---

## Round 0.3 · accepted · Accept

The authors addressed the reviewer's comments and the manuscript is now suitable for publication.